# Physical Simulation Test on Surrounding Rock Deformation of Roof Rockburst in Continuous Tunneling Roadway

Yaobin Shi [1,2,3], Yicheng Ye [1,2,3], Nanyan Hu [1,3,*], Yu Jiao [1] and Xianhua Wang [4]

1. School of Resource and Environmental Engineering, Wuhan University of Science and Technology, Wuhan 430081, China; shiyaobin@wust.edu.cn (Y.S.); yeyicheng@wust.edu.cn (Y.Y.); jiaoyu00wust@163.com (Y.J.)
2. State Environmental Protection Key Laboratory of Mineral Metallurgical Resources Utilization and Pollution Control, Wuhan University of Science and Technology, Wuhan 430081, China
3. Hubei Key Laboratory for Efficient Utilization and Agglomeration of Metallurgic Mineral Resources, Wuhan University of Science and Technology, Wuhan 430081, China
4. Sinosteel Wuhan Safety and Environmental Protection Research Institute Co., Ltd., Wuhan 430081, China; ssriwangxh@163.com
* Correspondence: hunanyan@wust.edu.cn

**Abstract:** To study the occurrence process, as well as the temporal and spatial evolution laws, of rockburst disasters, the roof deformation of continuous heading roadways during rockburst was studied through a physical similarity simulation test with a high similarity ratio and low strength. The deformation and failure evolution law of the roadway roof in the process of rockburst were analyzed by using detection systems, including a strain acquisition system and a high-power digital micro-imaging system. The results show that the rockburst of the roadway roof can be divided into four stages: equilibrium, debris ejection, stable failure, and complete failure stage. According to the stress state of a I–II composite crack, the theoretical buckling failure strength of the surrounding rock is determined as 1.43 times the tensile strength. The flexural failure strength of a vanadium-bearing shale is 1.29–1.76 times its compressive strength. With continuous advancement in the mining time, the internal expansion energy of the roadway roof-surrounding rock in the equilibrium stage continuously accumulates. The fracture network continuously increases, developing to the stable failure stage, with bending deformation, accompanied by continuous particle ejection until the cumulative stress in the failure stage increases, and the tensile state of the rock surrounding the roof expands radially into deep rock. A microscopic damage study in similar material demonstrated that the deformation of the roadway roof is non-uniform and uncoordinated. In the four stages, the storage deformation of the rock surrounding the roadway roof changes from small accumulation to continuous deformation, to the left (or deep rock). Finally, the roadway roof-surrounding rock becomes completely tensioned. The research results presented in this study provide a reference for the prediction and control of rockburst in practical engineering.

**Keywords:** rockburst; roadway; roof-surrounding rock; physical similarity simulation test; digital image processing (DIP) technology

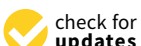

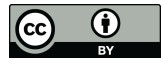

## 1. Introduction

In underground mining, roadway excavation causes the rock mass of the surrounding rock surface to change from a three-dimensional stress state to an approximately one- or two-dimensional stress state, resulting in stress readjustment and local concentration [1–3]. When the elastic strain energy stored in the rock mass exceeds the rock mass strength, the excess strain energy will be released violently, resulting in rockburst accidents, which pose serious threats to the safety of personnel [4–6].

In recent years, the frequency of rockbursts in mining underground and space engineering has increased significantly. The factors influencing rockburst are complex, mainly

including the excavation mode, surrounding rock stress state, rock mass structure, and its performance [7–11]. Scholars have deeply studied the incubation process, occurrence mechanism, and prevention technology of rockburst and have discussed the rockburst characteristics of rock masses in different environments [12,13]. In the study of rockburst mechanism, strength theory, stiffness theory, impact tendency theory, energy theory, instability theory, and fracture damage theory have been proposed [14]. The classification research has shown that there are more than five types of rockburst [15], and rock roofs with different thicknesses have distinct failure patterns under the complex stress [16]. Rockbursts can be divided into horizontal stress type, vertical stress type and mixed stress type, as well as burst ejection type, flake-spalling type and tunnel wall collapse type. Research has shown that rockbursts mostly occur in hard rock masses under high-stress environments, and that the severity is closely related to the stress concentration of the surrounding rock, which affects the ratio of rock-buckling failure strength ($\sigma_{\theta max}$) and rock uniaxial compressive strength, one of the important criteria for rockburst [17–19].

At present, research into rockburst has mainly been conducted through indoor rock mechanics testing, numerical analysis, field monitoring, and physical simulation testing. Physical similarity simulation testing is a main and important laboratory experimental method to study the development, expansion, rockburst, and final failure of rock fractures [20,21]. Jiang Lishuai et al. [22] have established two physical models utilizing similar materials and studied the influence of the advancing direction on the mining effect caused by a fault. Rockbursts affected by faults with different mining directions were compared and analyzed, and the correlation among advancing direction, strata behaviors, and rockburst induction was studied. Zhou Nan et al. [23] the used experimentally measured strain–stress curves of the backfill body and similarity theory in order to design and employ four experimental models for physical simulation. A non-contact strain measurement system and pressure sensors were used to monitor the deformation of the overlying strata and changes in abutment stress in front of the face during mining of the models for varying roof-controlled backfilling ratios. Chen Bing et al. [24] developed rock-like materials for simulating faults and surrounding rocks and carried out simulation experiments on the activation process of different types of analogical faults. They found that the failure process of rock-like samples with analogical concealed faults and analogical conduction faults could be divided into three stages, including the crack initiation, crack generation, and propagation stages. Li Shugang et al. [25] have carried out uniaxial compression experiments with different loading rates using a DYD-10 electronic universal testing machine for physical similarity simulation testing in order to study the influence of fracture evolution characteristics of rock-like materials under different strain rates. Their results show that, with an increase in the strain rate, the peak strength of the rock-like specimen increased, and the main fracture gradually developed in the direction parallel to the maximum principal stress, which led to the failure form of the specimen evolving from shear failure to tensile failure. Ma Dan et al. [26] studied non-Darcy hydraulic properties and the deformation behaviors of granular gangues through laboratorial, theoretical, and in situ aspects. They conducted a series of compression and seepage tests on granular gangues under the variables original grain size grade (GSG) and stress rate. They found that the fractal dimension reveals more obvious increases in the samples with the higher original GSGs and lower stress rates. Additionally, an improved model was established to predict the permeability evolution by the fractal dimension in their research.

Therefore, using physical simulation models bearing similarity to the field geometry and environment to simulate the rockburst process can more accurately reflect the dynamic rockburst development process. However, the occurrence process and temporal and spatial evolution law of rockbursts in high-stress roadways need to be further studied, as they have important reference significance for eliminating rockburst disasters and improving rockburst prediction and control.

In this paper, based on a rockburst-like material test featuring low strength and high brittleness in a gently inclined multi-layer deposit, a large-scale physically similar simula-

tion test model was constructed. The strip-mining method was adopted, and the strain characteristics and laws of roadway-surrounding rock under different mining intensity were analyzed using XL2101g multi-point high-speed whole process controlled static strain acquisition system and digital image processing (DIP) technology. The occurrence process and evolution law of rockburst in the rock surrounding the roadway—comprising a gently inclined multi-layer deposit—was studied, and the buckling failure strength and uniaxial compressive strength of the rock mass were determined, thus providing a scientific basis for revealing the temporal and spatial evolution law of rockburst.

## 2. Materials and Methods

### 2.1. Engineering Background

We studied the mining project of the Shanghengshan multi-layer shale deposit, which has 12 ore bodies in the ore section. The ore bodies are produced in layers, with good continuity and a simple shape. The ore-bearing rocks are mainly carbonaceous shale, siliceous shale, and vanadium-bearing shale, followed by siliceous rock. The ore body has a dip of 150–220° and an inclination of 5–25°. The ore length is 615–952 m, the thickness is 0.75–7.27 m, the thickness variation coefficient is 37.07–64.59%, and the inclined extension depth is 103–223 m.

The brittle rock shale with obvious rockburst tendency was chosen as the test material to establish the roadway rockburst model [27]. The physical and mechanical parameters of shale materials were obtained by uniaxial compression and Brazilian splitting tests, as detailed in Table 1.

**Table 1.** Physical and mechanical parameters of rock mass in Shanghengshan Mine.

| Rock Formation | Density /(kg/m³) | Porosity /% | Compressive Strength /MPa | Tensile Strength /MPa | Elastic Modulus /GPa | Poisson's Ratio |
|---|---|---|---|---|---|---|
| Vanadium-bearing shale | 2 482.53 | 14.33 | 76.69 | 12.56 | 58.80 | 0.21 |
| Carbonaceous shale | 2 429.86 | 18.61 | 49.45 | 7.42 | 50.80 | 0.20 |
| Siliceous shale | 2 564.72 | 16.07 | 112.63 | 17.24 | 59.80 | 0.21 |

### 2.2. Sample Preparation of Similar Materials

Considering the balanced equation, geometric equation, physical equation, stress boundary conditions, and displacement boundary conditions of the prototype and model, rock-like materials and rock materials followed the similarity criteria. In the design of the physical similarity stimulation test [28], we determined the geometric physical quantity similarity ratio as $C_l = 100$ and mechanical similarity ratios as $C_\rho = 1.5$ and $C_\sigma = 150$. The dimensionless parameters, such as Poisson's ratio, the porosity, and internal friction angle of similar materials were selected with the similarity constant of 1. Meanwhile, the geometric similarity constant and mechanical similarity constant met the following requirements.

$$C_\sigma / (C_\rho C_l) = 1 \tag{1}$$

Cylindrical standard samples with a diameter of 50 mm and a height of 100 mm were selected for samples of similar materials. Portland cement and gypsum were selected as the cementing materials, ordinary river sand as the filling material, and glycerol as the cementing agent. The proportion test of similar materials had been previously given in the literature [28]. The proportions and physical and mechanical parameters of similar materials in each mine and rock stratum of the multi-layer vanadium-bearing shale deposit were determined, as shown in Table 2, and the actual similarity constants of the similar materials are shown in Table 3.

### 2.3. Test Equipment and Methods

A self-developed adjustable similar simulation experimental device system [29] was adopted for the test, and the layered pressing method is used for model construction.

We determined the amount of similar materials for different rock layers of the physical similarity model according to the similarity ratio. We filled and tamped, layer by layer, from bottom to top, and used mica powder between the different rock layers (thickness 2–3 mm) to build a 4500 mm × 2200 mm × 400 mm (length × height × width) physical similarity simulation model, as shown in Figure 1.

**Table 2.** Similar material selection and mechanical parameters.

| Rock Formation | $\rho$ /(kg/m$^3$) | $\sigma_c$ /MPa | $\sigma_t$ /kPa | n /% | E /GPa | $\nu$ | c /kPa | $\phi$ /(°) |
|---|---|---|---|---|---|---|---|---|
| Siliceous shale | 1 704.41 | 0.75 | 114.87 | 16.21 | 0.399 1 | 0.213 3 | 71.218 | 41.087 |
| Vanadium-bearing shale | 1 661.74 | 0.51 | 84.01 | 14.46 | 0.394 4 | 0.210 0 | 61.891 | 40.275 |
| Carbonaceous shale | 1 620.98 | 0.33 | 49.42 | 18.56 | 0.336 8 | 0.203 3 | 56.748 | 40.106 |

**Table 3.** Similar constants of similar materials.

| Rock Formation | Density | Compressive Strength | Tensile Strength | Elastic Modulus | Poisson's Ratio | Cohesion Internal | Friction Angle | $C_\sigma/(C_\rho \cdot C_l)$ |
|---|---|---|---|---|---|---|---|---|
| Siliceous shale | 1.505 | 150.173 | 150.082 7 | 149.837 | 0.985 | 150.243 | 0.998 | 1.002 0≈1 |
| Vanadium-bearing shale | 1.494 | 150.373 | 149.506 0 | 149.087 | 1.000 | 150.264 | 1.003 | 1.000 3≈1 |
| Carbonaceous shale | 1.499 | 149.848 | 150.141 6 | 150.831 | 0.984 | 149.785 | 0.997 | 0.993 5≈1 |

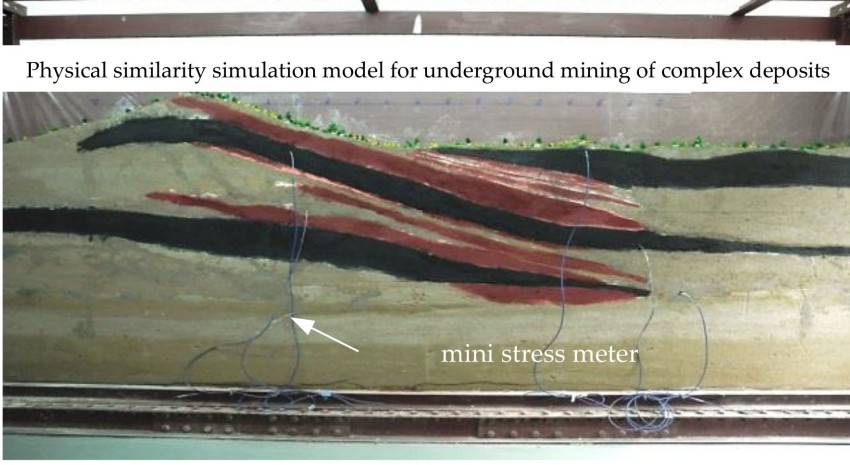

Physical similarity simulation model for underground mining of complex deposits

mini stress meter

**Figure 1.** Physical similarity simulation test model of gently inclined multi-layer deposit.

In the physical similarity simulation test, only the influence of rock self-weight stress was considered, and the influence of stress in other directions was not considered, such as in the radial direction.

In the physical similarity simulation test model, the XL2101G multi-point high-speed whole process controlled static strain acquisition system was used to monitor the vertical strain of the surrounding rock. The layout of sampling points for vertical strain is shown in Figure 2. In addition, a mini stress meter was used to detect the rock stratum pressure. The sample and instrument were consistent with the ambient temperature before the test, and we ensured that the laboratory was closed during the test. The mining of a certain sub rock stratum was carried out, and the layout of the ore room adopted a "mining one by one" approach in the simulated mining test [30]. The design width of pillar and stope was 15 m. To simulate the explosion process of the rock surrounding the roadway during roadway excavation, continuous mining without backfilling was selected for the test; that is, while maintaining the consistency of roadway section shape and size and the spacing distance between two adjacent roadways, the elastic strain energy of the simulated roadway-surrounding rock continues to accumulate and increase.

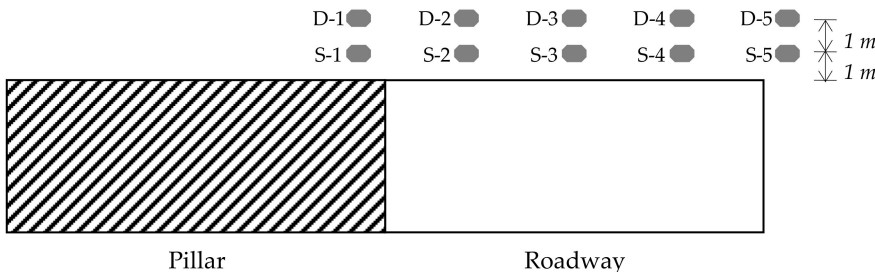

Pillar                    Roadway

**Figure 2.** Layout of sampling points of roadway roof.

Roadway excavation causes the stress of the rock mass in the surrounding rock surface to change from a three-dimensional stress state to an approximately one- or two-dimensional stress state, resulting in stress readjustment and local concentration. When the elastic strain energy stored in the rock mass exceeds the rock mass strength, the excess strain energy is released violently, resulting in a rockburst accident [31–33]. To simulate the rockburst failure phenomenon of a roadway excavation, the 300 times digital micro-imaging system was used in the test. The roadway roof cracks were captured by the digital micro camera, the images of cracks were numerically processed using Origin software (9.4.2.380), and the roadway roof deformation and failure process (DIP) was analyzed at the microscopic level.

Taking the initial step excavation roadway (roadway 1#) as the sample, the evolution process of roadway 1# roof rockburst in continuous excavation was analyzed. The schematic diagrams of the test prototype and the mining are shown in Figure 3 [28]. The actual situation of Sections 1 and 2 of the mining stope is shown in Figure 4.

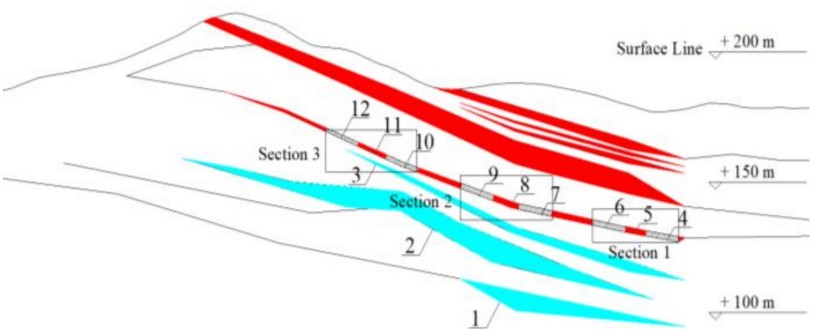

**Figure 3.** Schematic diagram of physical simulation test. Legend:1—Ore deposit V1; 2—Ore deposit V2 and V3; 3—Ore deposit V1; 4—Roadway1#; 5—Pillar 1#; 6—Roadway2#; 7—Roadway3#; 8—Pillar 2#; 9—Roadway4#; 10—Roadway5#; 11—Pillar 3#; 12—Roadway6#.

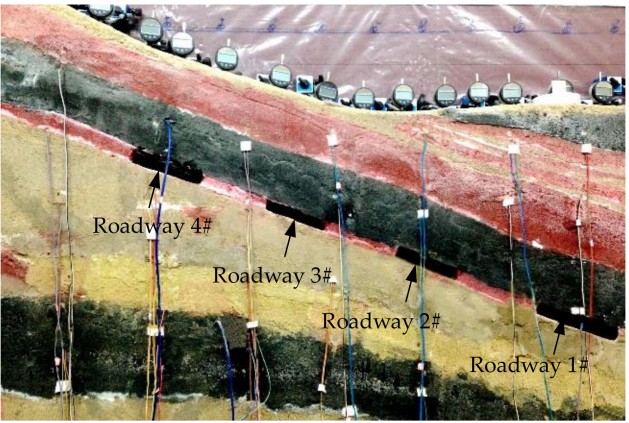

**Figure 4.** Status of similar simulation in mining Sections 1 and 2.

## 3. Stress Analysis of Roadway-Surrounding Rock Affected by Adjacent Face Mining

The main reason for the brittle fracture and even instability failure of rock is crack propagation and instability propagation throughout the rock [34,35]. The failure and fracturing of rock are essentially the same process; that is, the crack propagation in the surrounding rock leads to the final failure of rock, and the crack often shows the characteristics of mode I, mode II, or a composite mode [36,37].

### 3.1. Mechanical Model

Based on the theory of elastic–plastic mechanics [38,39], considering that the mining strip thickness of the deposit is smaller than the inclined length ($W_s$) of the working face and the strip pillar ($W_p$) interval, the working face can be calculated as an inclined crack, and the calculation and analysis model are shown in Figure 5.

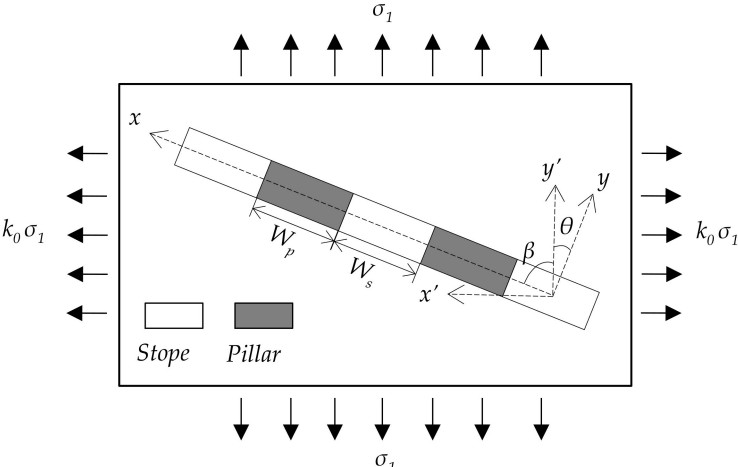

**Figure 5.** Plane stress mechanical model of roadway and pillar.

It was assumed that there is a penetrating oblique crack (strip width) with a length of $W_s$ in the infinite plate, and the edge is subjected to a uniformly distributed biaxial tensile pressure $\sigma_1$ and $k_0\sigma_1$ (where $k_0$ is the lateral pressure coefficient), where the vertical stress is $\sigma_1 = \gamma H$ ($\gamma$ is the unit weight of ore and rock; $H$ is the buried depth). Considering the working face direction and $\sigma_1$ the included angle of the action direction is $\beta$, and the edge is not under force.

### 3.2. Stress Function of Mode I and Mode II Crack Tip Affected by Mining in Adjacent Working Face

Cracks in the state of open mode I and sliding mode II were considered. For the two-dimensional elastic problems of mode I and mode II cracks, the Westergaard complex stress functions [40–42] for the tips of mode I and II cracks affected by the mining of adjacent working faces are given as Equations (2) and (3), respectively.

$$\left.\begin{array}{r} \sigma_x = \mathrm{Re}Z_\mathrm{I}(z) - y\mathrm{Im}Z'_\mathrm{I}(z) \\ \sigma_y = \mathrm{Re}Z_\mathrm{I}(z) + y\mathrm{Im}Z'_\mathrm{I}(z) \\ \tau_{xy} = -y\mathrm{Re}Z'_\mathrm{I}(z) \end{array}\right\} \tag{2}$$

$$\left.\begin{array}{r} \sigma_x = 2\mathrm{Im}Z_\mathrm{II}(z) + y\mathrm{Re}Z'_\mathrm{II}(z) \\ \sigma_y = -y\mathrm{Re}Z'_\mathrm{II}(z) \\ \tau_{xy} = \mathrm{Re}Z_\mathrm{II}(z) - y\mathrm{Im}Z'_\mathrm{II}(z) \end{array}\right\} \tag{3}$$

The tensor expression can be expressed by the following formula:

$$\sigma_{ij,\mathrm{I}} = \frac{K_\mathrm{I}}{\sqrt{2\pi r}} f_{ij,\mathrm{I}}(\theta) \tag{4}$$

$$\sigma_{ij,\Pi} = \frac{K_\Pi}{\sqrt{2\pi r}} f_{ij,\Pi}(\theta) \tag{5}$$

where $K_I$ and $K_{II}$ are the stress intensity factors of mode I and mode II cracks, respectively, $\theta$ is the crack angle, and $W_p = 2r$.

For the problem of mode I and mode II cracks in a an infinite plate, the stress intensity factor is defined as follows:

$$K_I = \lim_{|\zeta|=0} Z_I(\zeta) \cdot \sqrt{2\pi\zeta} \tag{6}$$

$$K_\Pi = \lim_{|\zeta|=0} Z_\Pi(\zeta) \cdot \sqrt{2\pi\zeta} \tag{7}$$

where, $\zeta$ is a new coordinate, with polar coordinates used to represent $\zeta = re^{i\theta}$.

When subjected to two-dimensional uniform tensile stress, the analytical function of the complex stress function of the through crack on the *x*-axis can be written as follows:

$$Z_I(\zeta) = \frac{\sigma(\zeta + a)}{\sqrt{\zeta(\zeta + 2a)}} \tag{8}$$

$$Z_\Pi(\zeta) = \frac{\tau(\zeta + a)}{\sqrt{\zeta(\zeta + 2a)}} \tag{9}$$

where $\tau$ is the uniform shear stress of the mode II crack problem (the main difference from the mode I crack problem), and $W_s = 2a$.

The stress intensity factor at the tip of an inclined crack on an infinitely thin plate is the product of the horizontal crack stress intensity factor and the correction coefficient, which is $F_I = k_0 \cdot cos^2\beta + sin^2\beta$ (where $F_{II} = (1-k_0) cos\beta \cdot sin\beta$). At the same time, considering the mutual influence between adjacent working faces, the stress at multiple inclined crack tips with certain spacing is the product of the stress at a single inclined crack tip and the correction coefficient, as shown in Equation (10) [43]. Therefore, bringing in Equations (6) and (7), the following formula is obtained:

$$\alpha = \left[ \frac{W_s + W_p}{\pi \frac{W_s}{2}} tg \frac{\pi \frac{W_s}{2}}{W_s + W_p} \right]^{1/2} \tag{10}$$

$$K_I = \alpha \cdot F_I \cdot \sigma \sqrt{\pi a} \tag{11}$$

$$K_\Pi = \alpha \cdot F_\Pi \cdot \sigma \cdot \sqrt{\pi a} \tag{12}$$

Bringing Equations (11) and (12) into Equations (4) and (5), respectively, we have:

$$\sigma_{ij,I} = \frac{1}{\sqrt{2}} \cdot \alpha \cdot F_I \cdot \gamma H \cdot \sqrt{\frac{W_s}{W_p}} f_{ij,I}(\theta) \tag{13}$$

$$K_\Pi = \alpha \cdot F_\Pi \cdot \sigma \cdot \sqrt{\pi a} \tag{14}$$

### 3.3. Stress Function at Crack Tip of I–II Composite Mode

The stress field at the tip of a mode I–II composite crack can be transformed into a superposition of mode I crack and mode II crack solutions; that is, the stress components near the tip of mode I and mode II cracks, as expressed in Equations (13) and (14), respectively, are superimposed. Therefore, the expression of the stress component at the crack tip under the of I–II composite mode is as follows:

$$\sigma_x = \frac{1}{\sqrt{2}} \cdot \gamma H \cdot \sqrt{\frac{W_s}{W_p}} \cdot \alpha \cdot \left[ F_I \cdot \cos\frac{\theta}{2} (1 - \sin\frac{\theta}{2} \sin\frac{3\theta}{2}) - F_\Pi \cdot \sin\frac{\theta}{2} (2 + \cos\frac{\theta}{2} \cos\frac{3\theta}{2}) \right] \tag{15}$$

$$\sigma_y = \frac{1}{\sqrt{2}} \cdot \gamma H \cdot \sqrt{\frac{W_s}{W_p}} \cdot \alpha \cdot \left[ F_{\mathrm{I}} \cdot \cos \frac{\theta}{2} (1 + \sin \frac{\theta}{2} \sin \frac{3\theta}{2}) + F_{\mathrm{II}} \cdot \sin \frac{\theta}{2} \cos \frac{\theta}{2} \cos \frac{3\theta}{2} \right] \tag{16}$$

For the plane strain problem, the maximum principal stress is as follows:

$$
\begin{aligned}
\sigma_{1,2} &= \frac{\sigma_x + \sigma_y}{2} \pm \sqrt{\left(\frac{\sigma_x - \sigma_y}{2}\right)^2 + \tau_{xy}^2} \\
&= \frac{1}{2\sqrt{2}} \cdot \gamma H \cdot \sqrt{\frac{W_s}{W_p}} \cdot \alpha \cdot \left[ F_{\mathrm{I}} \cdot \cos \frac{\theta}{2} - F_{\mathrm{II}} \cdot \sin \frac{\theta}{2} \cos \frac{\theta}{2} \cos \frac{3\theta}{2} \right] \\
&\pm \left\{ \left[ F_{\mathrm{I}} \cdot \cos \frac{\theta}{2} \sin \frac{\theta}{2} - F_{\mathrm{II}} \cdot \sin \frac{\theta}{2} \right]^2 + \left[ F_{\mathrm{I}} \cdot \sin \frac{\theta}{2} \cos \frac{\theta}{2} \cos \frac{3\theta}{2} + F_{\mathrm{II}} \cdot \cos \frac{\theta}{2} (1 - \sin \frac{\theta}{2} \sin \frac{3\theta}{2}) \right]^2 \right\}^{1/2}
\end{aligned}
\tag{17}
$$

For roadway-surrounding rock, we calculated $\sigma_{1,2} = 17.97 \pm 0 \, Mpa$.

According to the stress condition of a I–II composite mode crack, the crack is mainly affected by tensile and shear stress, and calculations show that $\sigma_{tmax} = 1.43\sigma_t$.

## 4. Results

### 4.1. Analysis of Roadway Roof Deformation

As shown in Figure 6, the cumulative stress state of the surrounding rock near the roadway roof was the same as that in the deep part of the roadway roof, while the stress concentration of the surrounding rock near the roadway roof was more obvious than that in the deeper part. The stress state of the surrounding rock on both sides of the roadway roof changed from a compression to a tension state, and the central monitoring point continued to be in a tension state. This indicates that maintenance of the roadway roof needs to focus on the rock ejection in the center of the roof in future mining work. The maximum strain or tensile position of the surrounding rock near the roadway roof is on the left side (cheek) of the working face, and the strain difference of the left-most monitoring point increased with the mining time step, consistent with the actual engineering phenomenon. This reflects that the monitoring point at the shallower part of the roadway roof along the rock stratum trend has greater pushing force than other parts. It can be concluded that the stress concentration of the surrounding rock is closely related to the strike of the rock stratum until rockburst. In other words, rockburst is closely related to the direction of tectonic stress [44,45].

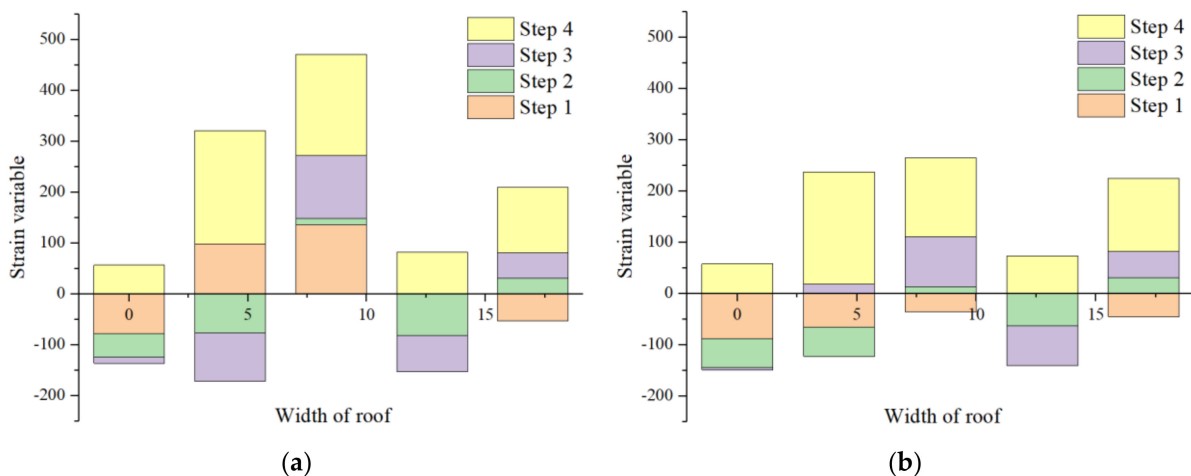

**Figure 6.** Strain of surrounding rock of roadway roof in different mining time step: (**a**) Monitoring points near roadway roof and (**b**) Monitoring points in the deep part of roadway roof.

In addition to geological factors, mining conditions and construction procedures are also very important factors affecting rockburst. The stress state of deep monitoring points of the roadway roof changed from a compression to a tension state with the advance of mining time. In the process of continuous mining, the roof of the adjacent roadway and the deep monitoring points of the roadway roof were in tension at the fourth stope step. This

indicates that with the continuous increase in the number of adjacent stopes, the dynamic load of the roadway roof and the surrounding rock stress of the first stope adjusted rapidly, which can easily lead to the occurrence of rockburst accidents.

### 4.2. Deformation and Failure Process of Roadway Roof

The data were graphically processed using Origin software, as shown in Figure 7. The analysis showed that the failure of the roadway roof could be roughly divided into equilibrium, debris ejection, stable failure, and complete failure stages.

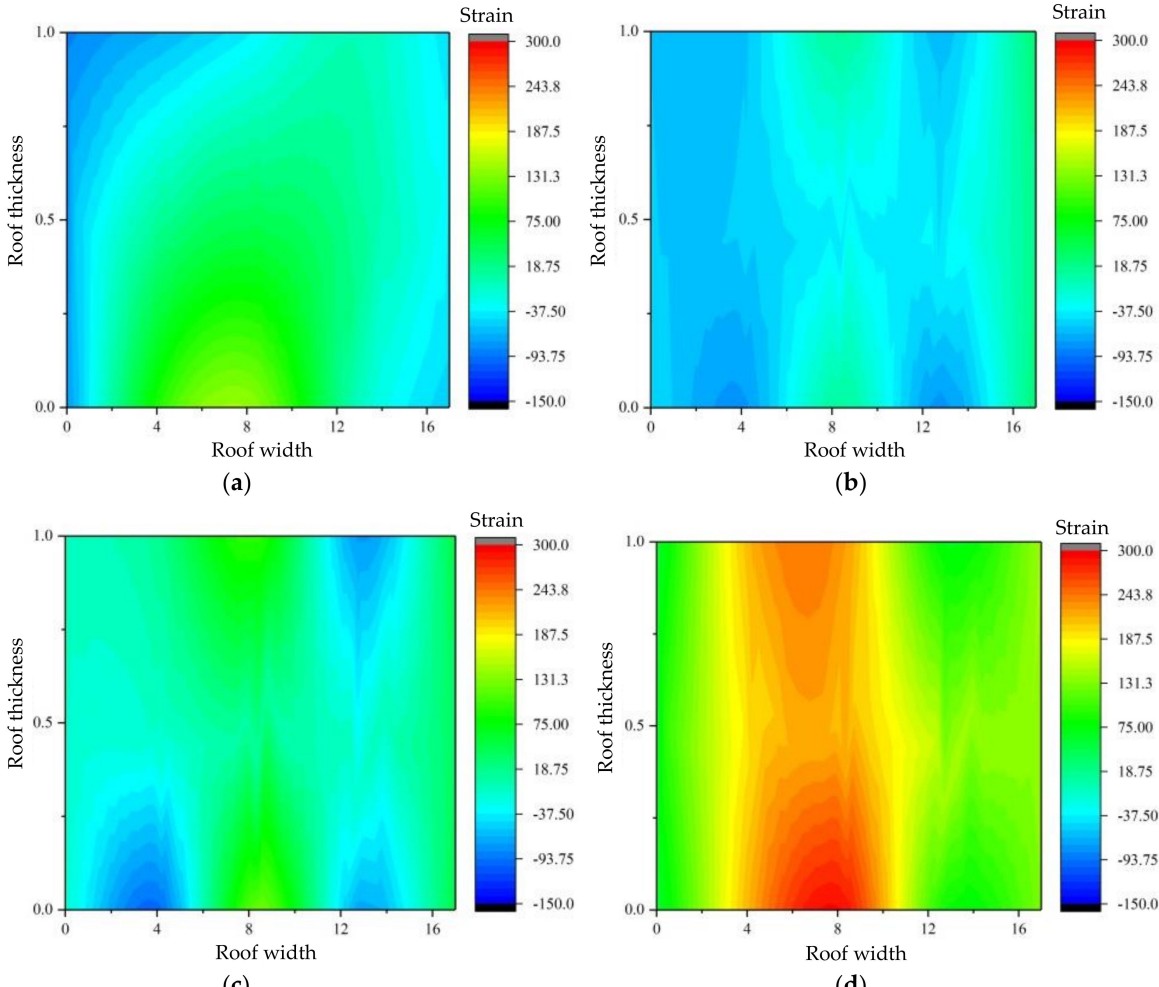

**Figure 7.** Deformation and failure process of roadway-surrounding rock: (**a**) First mining step; (**b**) Second mining step; (**c**) Third mining step; (**d**) Fourth mining step.

After the first mining step of roadway excavation, the overall deformation of the roof was small. The deformation of the roadway roof was pulled along the roadway centerline and distributed symmetrically. The surrounding rock near the roadway roof was in a certain tensile state, and the roadway roof was in a certain stable state.

With the continuous expansion of mining time step, in the second mining step, the roadway 1# midline roof was stretched, and deformed, the effects of which extended to the deep rock level. Meanwhile, adjacent monitoring points and the roof on both sides of the roadway were compressed to a certain extent. The roadway roof was in a non-uniform and uncoordinated deformation state, debris fell intermittently on the inner wall of the roadway, and roadway roof entered the debris ejection stage.

In the third mining time step, the overall tensile deformation of the roadway 1# continued to increase, during which the deep monitoring point on the left side of the

roadway (1.5 m from the roadway) changed from a compression to tensile state, and the compressive strain of the monitoring point near the roadway (0.5 m from the roadway) accumulated. The debris falling from the roadway roof continued to increase, and the roadway roof entered the debris ejection stage.

In the fourth mining time step, the roadway 1# roof was in a fully tensioned state, and the deformation increased sharply. The maximum tensile strain accumulation area shifted to the left side of the roadway, and the a tension state of the monitoring points near the roadway was obvious, compared with the deep monitoring points. On the model, it can be seen that the rock stratum on the surface of the model had formed fractures, the fracture of the roadway roof had a large expansion degree, and the expansion and connection of different fracture points formed a dense fracture network. Cracking damage had occurred on the left inner wall of the roadway and adjacent pillars.

### 4.3. Temporal and Spatial Evolution Law of Microscopic Damage and Failure in Roadway Roof Rockburst Process

Although the XLl2101G multi-point high-speed whole process controlled static strain acquisition system could observe the macro deformation of the roof of the continuous excavation roadway, the damage in the rock mass was not well reflected using this system. Digital image processing (DIP) technology can accurately characterize the microscopic damage and failure law in rock [46,47]. By analyzing the temporal and spatial evolution law of rock damage fracture extension, expansion and nucleation, the microscopic damage process of the roof slate rockburst in the continuous excavation roadway was studied using DIP technology.

DIP technology studies the temporal and spatial evolution law of microscopic rock damage and failure and shows the mathematical significance of microscopic rock damage. The images were collected by a 300 X digital microscopic imaging system, and the micro damage data of the sampled images were analyzed using the MATLAB software (7.10).

In the process of determining the micro damage data of the rock mass, it was necessary to determine the cross-scale transformation threshold from the collected image macro data to the micro damage micro data [48].

The gray value calculation method was used to determine the micro data of rock mass fractures. Through normalization of the sampling data from the five monitoring points in the first mining time step, the micro data of rock mass fractures were accumulated and calculated, respectively, according to $F$ = (0.05, 0.10, 0.15, . . . , 0.90, 0.95). Combined with the polynomial fitting of fracture and non-fracture data, $F$ = 0.35 was determined as the cross-scale transformation threshold from macro to micro data of rock mass damage, as shown in Figure 8.

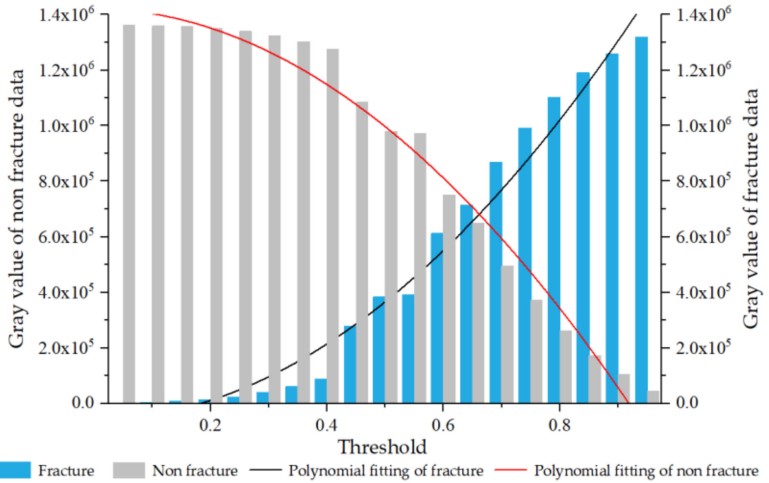

**Figure 8.** Determination of cross-scale transformation threshold from macro data to micro data of rock mass damage.

Taking the roof monitoring point S-3 of the roadway centerline as an example, microscopic image data collected at the sampling points in the first to fourth mining time steps are shown in Figure 9.

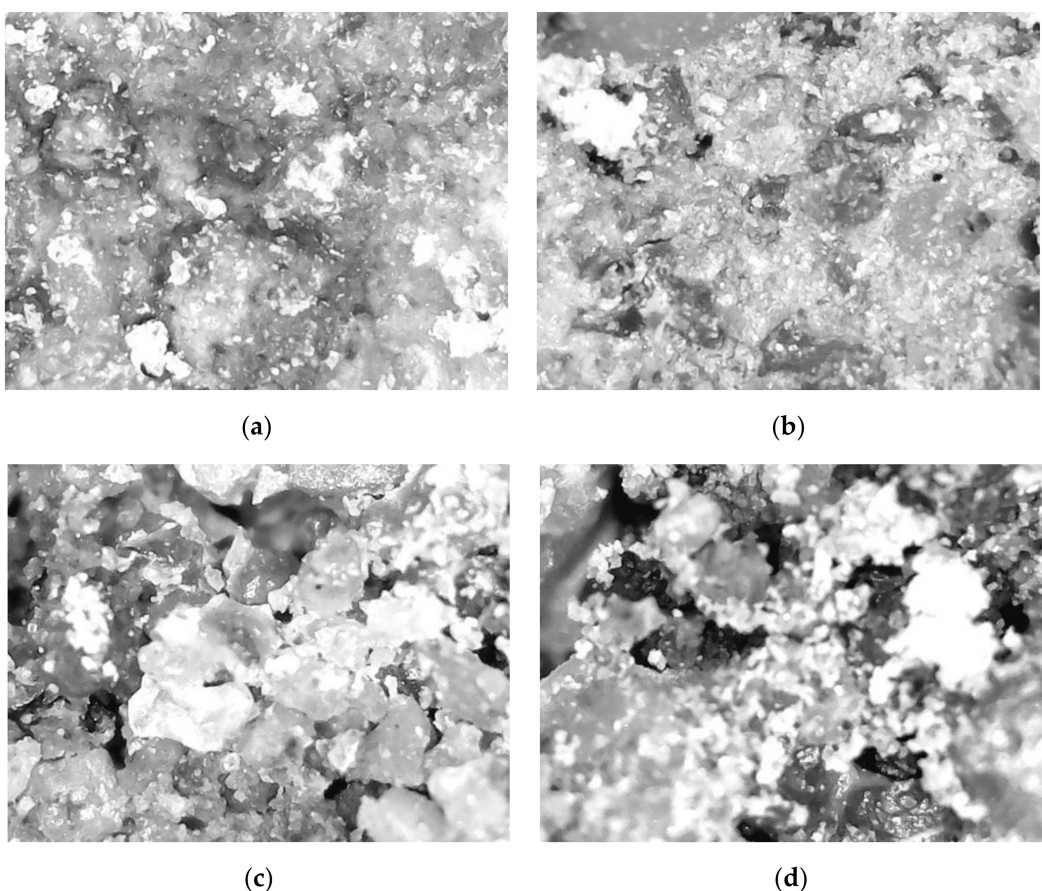

**Figure 9.** Images were acquired at different time steps at S-3 sampling points (300X): (**a**) First mining step; (**b**) Second mining step; (**c**) Third mining step; (**d**) Fourth mining step.

The microscopic images were converted into corresponding gray images and the Itophat through MATLAB to obtain the corresponding binary image, where the gray threshold change was used to realize the binarization of the binary image (Figure 10).

The binary images of monitoring points were obtained by numerical analysis. Black pixels in the image represent the cracked areas, while white pixel represent non-cracked areas. The data analysis showed that with the continuous advancement of mining time steps, the rock fractures in the S-3 roof increased, and the rock micro damage occurred through local crack occurrence and propagation, damage compaction, crack nucleation and extension and fracture network formation. After the second time step mining, the in situ stress in the post-mining rock promoted the compaction and closure of micro cracks in the surrounding rock, and the development and penetration of cracks was limited, which reduced the binary data of cracks; however, the cracks gathered and nucleated in the rock in the third time step. The expansion speed of rock fractures become faster, and the phenomenon of fracture nucleation in the new area was prominent. The reason for this is that, with the continuous progress of mining, the potential energy accumulated in the equilibrium stage increases significantly. After entering the debris ejection stage, the energy continues to accumulate and, at the stable failure stage, the energy release is concentrated, the process of fracture nucleation, and extension is accelerated. Finally, the fracture network is dense in the complete failure stage.

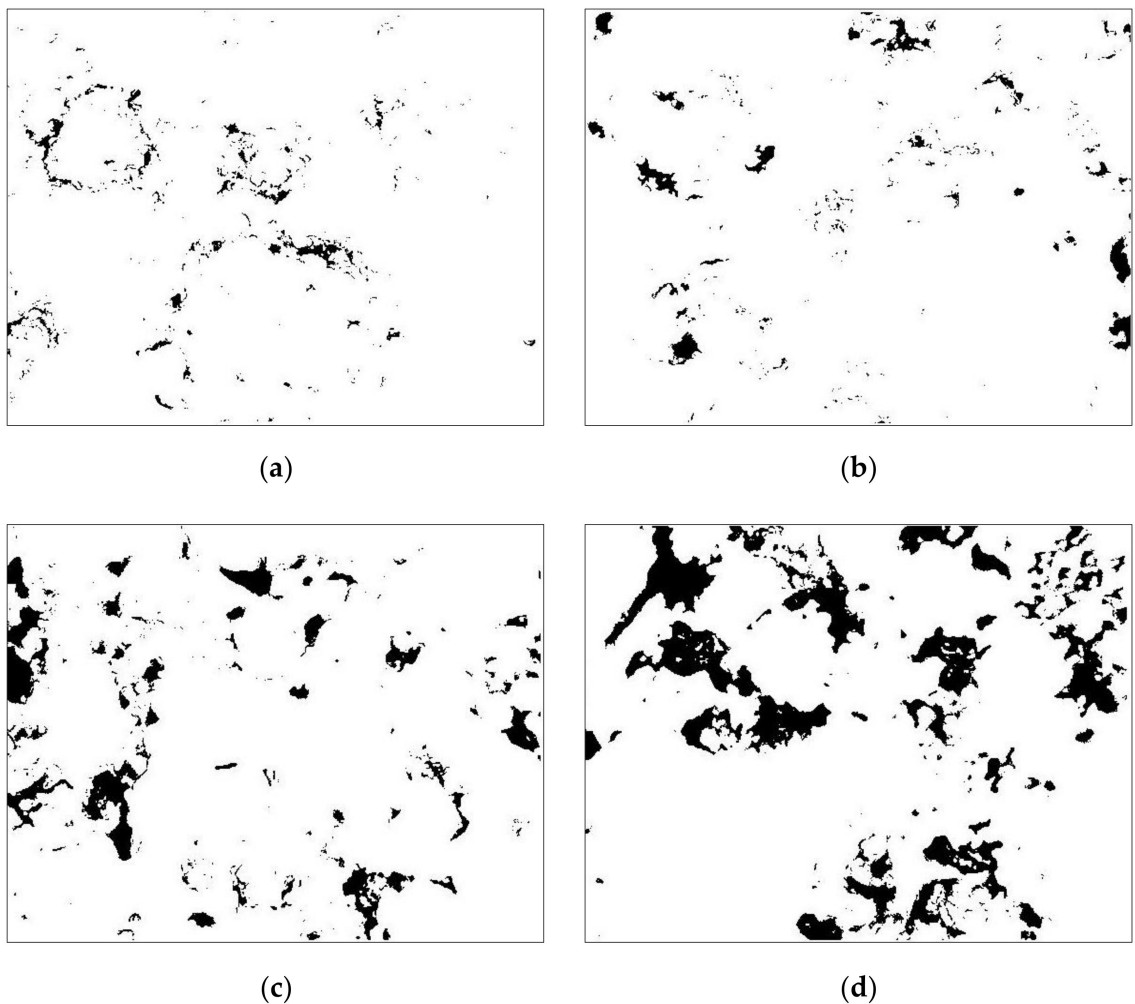

**Figure 10.** Gray binary image of S-3 sampling points (300X): (**a**) First mining step; (**b**) Second mining step; (**c**) Third mining step; (**d**) Fourth mining step.

According to the binary data, the density damage of S-3 sampling points in the first to fourth time steps was 5.75%, 2.64%, 5.98%, and 6.55%, respectively. The sampling points went through the process of crack expansion, compaction, and re-expansion. The gray level of the binary data of the sampling points was divided into high and low levels, as shown in Table 4. With a higher proportion of low gray level, more cracks are derived.

**Table 4.** Gray level distribution at different time steps of S-3 sampling points.

| Time Step | Low Gray Level/ Level 0–85 | Medium Gray Level/Level 86–170 | High Gray Level/ Level 170–256 |
|---|---|---|---|
| First | 0.93% | 44.97% | 54.10% |
| Second | 1.84% | 34.41% | 63.75% |
| Third | 5.37% | 35.85% | 58.78% |
| Fourth | 11.71% | 37.01% | 51.28% |

We analyzed the sampling data of the roof of the continuous driving roadway 1 # to determine the rock density damage according to the binarization diagram of sampling points at different time steps. By analyzing the micro damage change, we could determine the density damage increment ($\triangle$F) of the roof rock in different mining steps, as shown in Figure 11.

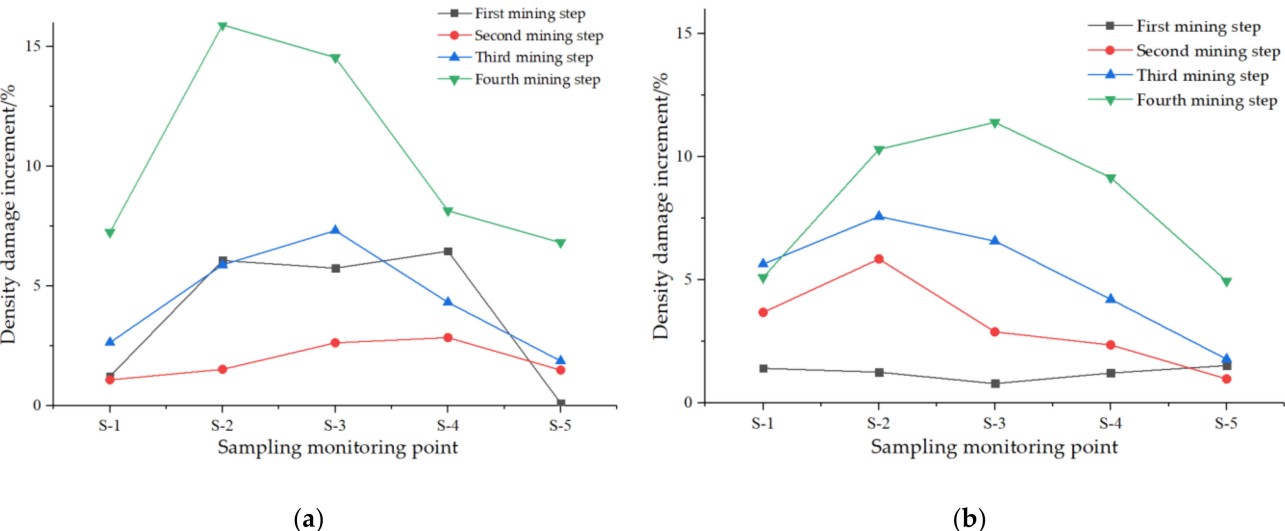

**Figure 11.** Density damage of roadway roof in (**a**) near roadway roof sampling points and (**b**) deeper sampling points.

The rock density damage of the roadway roof continued to increase with the advance of mining time, and the rock density damage increment at deeper sampling points continued to increase as a whole. In the equilibrium stage, the load of the rock surrounding the roadway roof was low, the rock density damage was symmetrical, and the density damage of surrounding rock near the roadway roof was high. In the stage of debris ejection, the density damage increment of the rock surrounding the roadway roof was less than that in the previous step, and the rock mass was in the stage of compaction and density. The density damage of the surrounding rock of the deep roadway roof continued to increase. In the stable failure stage, the density damage of the rock surrounding the roadway roof on the same vertical line was similar overall, whereas the density damage increment of left surrounding rock was larger than that of right surrounding rock. In the stage of complete failure, the density damage increment of the rock surrounding the roadway roof increased greatly; it was 1.5–4.9 times higher than that in the initial time step (abnormal points were eliminated). The density damage increment of the surrounding rock of the deep roadway roof was centrally symmetrical and evenly distributed, increasing by 2.5–7.2 times, when compared with the initial time step (abnormal points were eliminated), which explains the continuous extension process of the failure state of the roadway roof from the adjacent surrounding rock to the deep surrounding rock.

Using the density damage increment research method, the temporal and spatial evolution law of micro damage and failure in the process of roadway roof rockburst was analyzed, and the cumulative process of roadway roof deformation and temporal and spatial laws of failure were explained. The corresponding conclusions are similar to those drawn in Section 2.1.

*4.4. Deformation and Failure Characteristics of Roadway Roof*

Assuming that the surrounding rock of the rectangular roadway is regarded as a homogeneous and isotropic continuous medium, the conformal variation can be used to map the outside of a rectangular roadway to the unit circle through the mapping function $Z = \omega(\xi)$, as shown in Figure 12. Based on the deformation characteristics of rock and the theory of elasticity, the mechanical constitutive equation of the volume deformation of the roadway-surrounding rock is determined, with the following partial tensor form [49]:

$$\begin{cases} \left(D + \frac{k_1}{\eta}\right)\sigma_{rs} = \left[(k_1 + k_2)D + \frac{k_1 k_2}{\eta}\right]\varepsilon_{rs} \\ \sigma = 3k\varepsilon \end{cases} \tag{18}$$

where, $D$ is the differential operator for time ($t$), $D^n = \partial^n / \partial t^n$. $\sigma_{rs}$ is the partial stress tensor, $\varepsilon_{rs}$ is the strain partial tensor, $k_1$ is the shear deformation modulus of viscous components of the surrounding rock, $k_2$ is the shear deformation modulus of the elastic components of the surrounding rock, and $\eta$ is the viscosity coefficient of the viscous component of the surrounding rock.

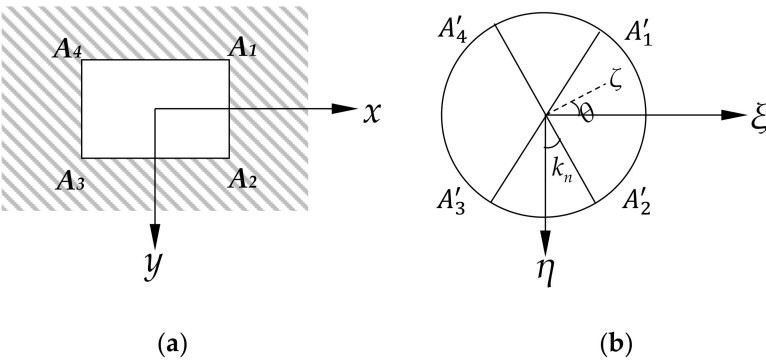

<div align="center">(a)</div><div align="center">(b)</div>

**Figure 12.** Mapping of Z plane rectangle (**a**) to $\zeta$ plane unit circle (**b**).

It can be found that the stress tensor of roadway-surrounding rock is positively correlated with the strain tensor and is not related whatsoever with the process of reaching this state.

The stress accumulation increment of the rock surrounding the roadway roof is shown in Figure 13. With the initial excavation, the shear stress of the rock surrounding the roadway roof gradually concentrated, and the stored energy is small. After entering the debris ejection stage, the strain energy in the surrounding rock accumulated continuously, small particles were ejected from the surrounding rock, the ejected particle size gradually increased, and there was no obvious buckling deformation of the roof. However, the high-power microscopic imaging system showed that tensile crack failure had begun to occur in the surrounding rock, forming new tensile cracks, which continued to expand and nucleate. When it developed to the stable failure stage, under the further action of tangential stress, the elastic strain energy accumulating in the surrounding rock of the roof adjacent to the roadway continued to increase, the deformation state of the surrounding rock in the deep part of the roof around the roadway centerline obviously changed, and the local bending deformation of the surrounding rock was accompanied by continuous particle ejection. The high-power microscopic imaging system showed that the surrounding rock fractures expanded intensively, forming a very obvious fracture network (or fracture family). When mining in the fourth step, the cumulative stress in the surrounding rock roof surged, the tensile state of the roadway centerline roof expanded to the deep rock along the radial direction, the cumulative deformation curve formed a V-shaped rockburst zone, and the deformation deflection tensor of monitoring points near the corner of the low-level (high burial depth) roof changed from a compression to a tension state.

According to the Hoek brown rock mass strength failure criterion [50]:

$$\sigma_1 = \sigma_3 + \sigma_{\text{ci}} \left( m_b \frac{\sigma_3}{\sigma_{cj}} + s \right)^{0.5} \tag{19}$$

where, $\sigma_1$ is the maximum principal stress when the rock mass is damaged($MPa$), $\sigma_3$ is the minimum principal stress in the case of rock mass failure($MPa$), $\sigma_{ci}$ is the rock block strength($MPa$), and $m_b$ and $s$ are dimensionless parameters representing the properties of rock materials, which are taken as 10 and 1, respectively, in this study.

Considering that the inner wall of the roadway after excavation showed $\sigma_3 = 0$, the roadway buckling failure satisfies $\sigma_{\theta max} = \sigma_1 = \sigma_{ci}$; that is, in theory, when the tangential stress of surrounding rock around the roadway reaches the uniaxial compressive strength,

the roadway will be damaged. However, in our experimental study, we found that the buckling failure strength of the roof from the first to third time steps was $\sigma_{\theta max} \approx (1.29–1.76)$ $\sigma_{ci}$, as shown in Table 5. This is also consistent with previous research results at home and abroad. The results of the analysis are due to the influence of the size effect and the strong effect on the roof after roadway excavation [35,51,52].

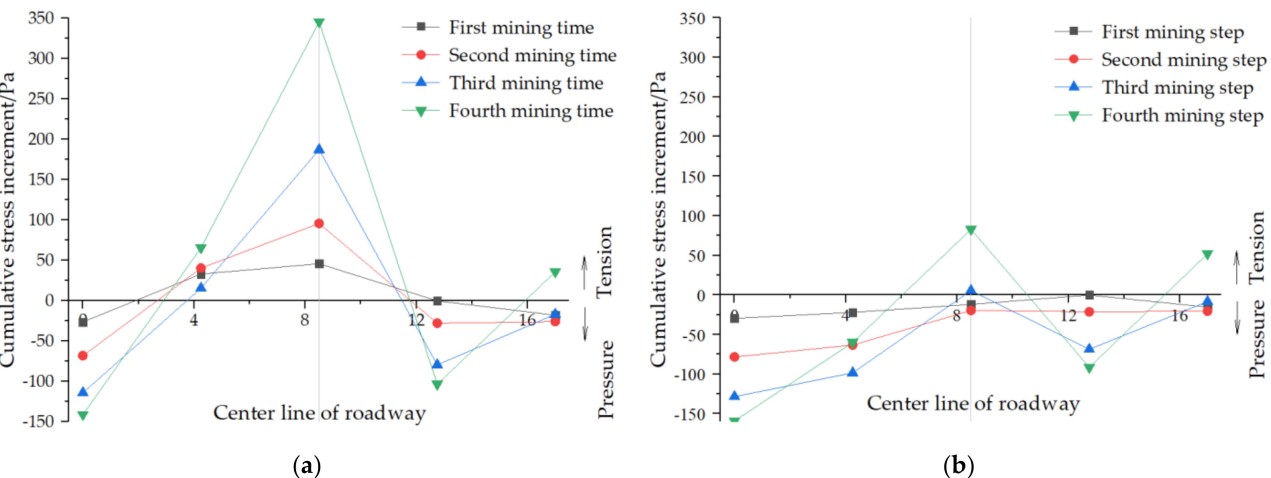

**Figure 13.** Cumulative stress increment of roadway roof-surrounding rock: (**a**) Near roadway roof sampling points and (**b**) deeper sampling points.

**Table 5.** Buckling strength coefficient of roadway roof-surrounding rock.

| # | D-1 | D-2 | D-3 | D-4 | D-5 |
|---|---|---|---|---|---|
| Peak load/Pa | 255.630 5 | 225.563 7 | 146.756 4 | 195.853 1 | 147.642 0 |
| $\sigma_{\theta max}/\sigma_c$ | 1.57 | 1.29 | 1.69 | 1.35 | 1.59 |
| # | S-1 | S-2 | S-3 | S-4 | S-5 |
| Peak load/Pa | 241.386 9 | 40.3965 8 | 187.357 7 | 206.746 4 | 153.157 1 |
| $\sigma_{\theta max}/\sigma_c$ | 1.68 | 1.31 | 1.76 | 1.44 | 1.63 |

## 5. Discussion

In this study, special emphasis was put on developing simple theoretical tools that could rapidly assess the stability of the surrounding rock for a continuous driving roadway. Such simplicity and efficiency are crucial for practicing engineers, in particular, during indoor physical similarity simulation and rock mechanics test design. In the laboratory test, some parameters of rock materials were idealized, ignoring uncertain factors such as rock hydrogeology, natural weathering, and disturbance. In view of the surrounding rock deformation and failure process of underground engineering under the action of the overburden of self-weight and support pressure in geotechnical engineering, the following assumptions were made [53]:

1.  The rock material studied is a brittle material, which can be considered to have linear elasticity and can be studied using the relevant theories of linear elastic fracture mechanics;
2.  The rock material was considered to be isotropic;
3.  The deformation and failure process of the surrounding rock is quasi-static and isothermal;
4.  In the micro research of rock, the cracks studied are regarded as ideal cracks, and the crack size is much smaller than the rock mass size. Ignoring the influence of finite size on the calculation results, it is assumed that the rock mass size is infinite.

Whether the judgment of the initial in situ stress field is reliable, and whether the selection of rock mass parameters is reasonable, will directly affect the economy, reliability, and safety of engineering design and construction [54]. E. T. Brown and E. Hoek [55] summarized the measurement results of in situ stress in different regions of the world in

1978 and summarized the vertical stress in countries all over the world, $\sigma_v$, and changes with the depth, H, with the fitting formula as follows.

$$\sigma_v = 0.027H \qquad (20)$$

The ratio of average horizontal in situ stress to vertical in situ stress was obtained, and the relationship is as follows.

$$\frac{100}{H} + 0.30 \leq k_0 \leq \frac{1500}{H} + 0.50 \qquad (21)$$

In this paper, the regression analysis equation of Zhao D.A. for the ratio of average horizontal in situ stress to vertical in situ stress of sedimentary rocks in China was selected [56], as is shown in Figure 14.

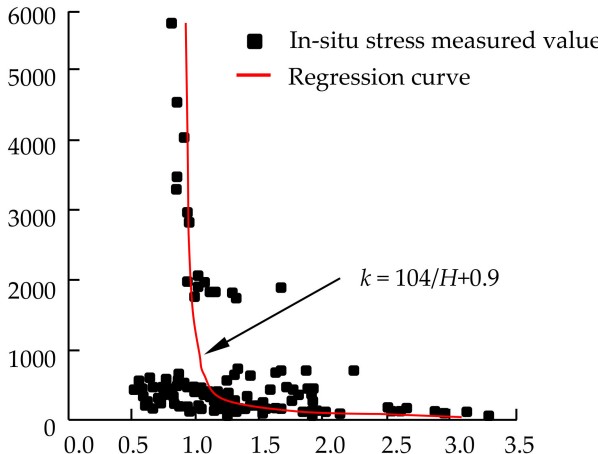

**Figure 14.** Variation law of average horizontal in situ stress and vertical in situ stress ratio with the depth of sedimentary rocks in China.

At the same time, the comparative analysis of the rock flexural failure strength coefficient, as calculated in Section 2, showed that there was a certain gap between the rock flexural failure strength coefficient obtained by theoretical analysis and the maximum flexural failure strength coefficient calculated by physical similarity simulation experiment. Affected by the characteristics of multi-layer strata, mining disturbances, and monitoring of time effects, the compaction of the roadway-surrounding rock roof occurred to a certain extent, and its endurance strength was enhanced in the process of the physical similarity simulation test. Moreover, due to the large geometric similarity selected in the physical similarity simulation test, there must be a certain error between the stress moment of the roof of the overhead roadway and the actual project.

Correspondingly, from the experimental results, the average flexural failure strength coefficient of 10 sampling monitoring points was 1.531, which differed by 7.01% from the rock flexural failure strength obtained by theoretical analysis. The theoretical flexural failure strength coefficient was still within the range of the experimental research results, and the error met the requirements of test regulations. Therefore, we believe that the comparison results can still reasonably approximate the theoretical analysis results and reflect the final results of rock deformation and failure.

## 6. Conclusions

In this study, we used a physical similarity simulation test to carry out a roof rockburst simulation test considering underground roadway excavation. The incubation process of rockburst was studied using XL2101g multi-point high-speed whole process controlled static strain acquisition system and digital micro-imaging system. The evolution law of roof

deformation during roadway excavation was preliminarily revealed. Our main conclusions are as follows.

1. Failure of the roadway roof can be divided into four stages. In the equilibrium stage, the expansion energy in the roof accumulates continuously, accompanied by the continuous increase in the fracture network. At the stage of stability failure, with the deep surrounding rock of the roof around the roadway centerline as the center, the local bending deformation of the surrounding rock of the roof appears, accompanied by continuous particle ejection. Finally, in the failure stage, the roof stress increases accumulatively, the tensile state of the roof of the roadway centerline expands to the deep rock along the radial direction, the cumulative deformation curve forms a V-shaped rockburst zone, and the surrounding rock deformation is obvious.

2. The deformation of roadway roof showed non-uniform and uncoordinated deformation. In the equilibrium stage, the stored deformation energy of the roadway roof is small, and the roof deformation is centrosymmetric. After entering the clastic stage, the strain energy of the surrounding rock accumulates continuously, small particle ejection occurs in the surrounding rock, and the strain energy of the surrounding rock near the corner of roadway increases continuously. In the stable breaking stage, the surrounding rock around the roadway centerline is deformed obviously, and local buckling deformation occurs, forming an obvious fracture network (or fracture cluster). In the failure stage, the stress concentration of the roadway roof shifts to the left side of the roadway, and the roadway is fully tensioned as a whole.

3. The rock surrounding the roadway roof is mainly affected by tensile stress and shear stress. According to the stress state of a I-II composite crack, it was determined that the theoretical buckling failure strength of the roadway roof-surrounding rock is 1.43 $\sigma_t$. The results of the physical similarity simulation model test show that the theoretical buckling failure strength of the rock surrounding the roadway roof is (1.29–1.76) $\sigma_{ci}$. The analysis results are due to the influence of the size effect and the strong effect on the roof after roadway excavation. We expect that the roof will be affected by size and strength effects after roadway excavation. The theoretical flexural failure strength coefficient was still within the range of the experimental research results, and the error met the requirements of test regulations.

**Author Contributions:** Conceptualization, Y.Y., Y.S., and X.W.; methodology, N.H., Y.Y. and Y.S.; software, Y.S.; validation, Y.S. and Y.J.; formal analysis, N.H., Y.Y. and Y.S.; investigation, N.H. and Y.S.; resources, Y.Y. and N.H.; data curation, Y.S. and Y.J.; writing—original draft preparation, Y.S., Y.Y., and N.H.; writing—review and editing, N.H., Y.S. and Y.Y.; visualization, Y.S.; supervision, N.H. and Y.Y.; project administration, Y.Y. and X.W.; funding acquisition, Y.Y. and Y.S. All authors have read and agreed to the published version of the manuscript.

**Funding:** This research was funded by the Natural Science Foundation of Hubei Province (No.2020CFB123), Scientific Research Project of Hubei Provincial Department of Education (No.Q20201109), Key Research and Development Plan of Hubei Province (No. 2020BCA082), and Innovation and entrepreneurship training program for college students of WUST (No. 20ZB012).

**Institutional Review Board Statement:** Not applicable.

**Informed Consent Statement:** Not applicable.

**Data Availability Statement:** Not applicable.

**Conflicts of Interest:** The authors declare no conflict of interest.

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
