# Peer review of "Physical Simulation Test on Surrounding Rock Deformation of Roof Rockburst in Continuous Tunneling Roadway"

_minerals, doi:10.3390/min11121335_

Round 1
Reviewer 1 Report
This paper has made a useful attempt to investigate the surrounding rock deformation during roof rockburst, with laboratory test and mechanical analysis. Overall, this work is interesting, and the authors made lots of efforts on the strain filed and microscopic roadway roof. However, the paper is marred by rather poor descriptive or clumsy wording including a number of typographical errors and even incorrect grammars. It is really difficult for the reviewer to read, and therefore the English language must be improved by an expert who is good at English. Thus, I would like to recommend major revision of the manuscript considering a few suggestions listed below.
- The title is confusing. It is better to change it to ʹʹ Physical Similarity Simulation Test on Surrounding Rock Deformation of Roadway during Roof Rockburstʹʹ
- Page 1, line 18, ʹʹ …through high similarity ratio and low strength physical similarity simulation test. ʹʹ is grammatically incorrect. Please change it to ʹʹ…through physical similarity simulation test with high similarity ratio and low strength. ʹʹ
- Page 1, line 24, what is the meaning of σt? The authors should note the meanings of symbols in the manuscript where they first appear.
- Page 1, line 30, what is like-rock microscopic damage study? Like-rock or rock-like?
- There is some problems with the structure of this manuscript. The authors should introduce the engineering background first before the experimental materials and setup. Also, the in-situ stress where the roadway locates should be illustrated as well as the loading method of the physical similarity simulation test equipment. One-dimensional loading or two-dimensional loading?
- Page 4, lines 134-135, the sentence lacks subject. Please correct it.
- In section 2.1, the similarity ratio of all parameters should be explained, such as stress, density and size.
- The authors stated that the strain field during the mining process can be obtained by the advanced XL2101G multi-point high-speed whole 131 process controlled static strain acquisition system. Does the strain refer to the horizontal strain, the vertical strain or the maximum principal strain? Besides, the reviewer cannot determine where the roof of the cloud diagram is in the roadway. It is better to add some sentences to describe the location of the shown roof.
- In section 3, What problems can the mechanical model solve? What is the relationship between this part and other chapters?
- How to judge the occurrence of rockburst based on the physical similarity simulation test results?
- For the convenience of readers, the conclusions should show the serial number of each point.
- The introduction of this manuscript is not complete enough for the mining crack evolution and rock burst. Therefore, the introduction needs to be supplemented.
- Examples of typos or grammatical errors are as follows:
Page 1, line 17, delete the second ʹʹroofʹʹ.
Page 1, line 42, ʹʹone-way or two-wayʹʹ should be ʹʹone-dimensional or two dimensional ʹʹ.
Page 2, line 86, delete “we”.
Page 3, line 124, ʹʹDetermineʹʹ should be ʹʹDeterminingʹʹ.
Page 4, line 131, ʹʹXL2101G multi-point high-speed whole 131 process controlled static strain acquisition systemʹʹ should not be used in italics, similarly hereinafter.
Page 4, line 133, ʹʹearth pressure boxʹʹ should be ʹʹmini stress meterʹʹ
Page 15, line 429, ʹʹDiscussʹʹ should be ʹʹDiscussionʹʹ.
Reviewer 2 Report
I would like to thank the authors for completing the manuscript in such a wonderful way, and in my opinion, it is acceptable.
Author Response
Thank you for your letter and for the reviewers’ comments concerning our manuscript entitled “Physical similarity simulation test on Surrounding rock deformation before and after roof rockburst in continuous tunneling roadway”. The revised portions are marked up using the “Track Changes” function in the manuscript.
Once again, thank you very much for your comments and suggestions.
Reviewer 3 Report
The article is well written. The individual chapters are logically related to each other. The greatest advantage of the article is laboratory research, which is very interesting both from a scientific and technological point of view. Below are some comments and suggestions:
- For Figure 3, a few sentences should be added in the text regarding the scale presented on the model and how the connection between the individual layers was modeled.
- For the sentence in line 170 "Based on the theory of elastic-plastic mechanics", at least one literature should be provided.
- Line 415, Hoek Brown entry should be corrected.
- In the Chapter 4.3 it should be written how the models for research were prepared using Digital image processing technology.
- In the chapter on conclusions, it should be made clearer how the research results can be used for industry.
- Minor errors such as the distance between the number and the unit should be corrected; lines: 111, 126, 127.
Round 2
Reviewer 1 Report
The authors have addressed my comments well. However, the last two comments can be improved.
(1) The title is wordy, it is suggested to be changed as: Physical simulation test on surrounding rock deformation of roof rockburst in continuous tunneling roadway.
The introduction of this manuscript is not complete enough for the rock crack evolution. Therefore, the introduction needs to be supplemented, e.g., (â…°) Reutilization of gangue wastes in underground backfilling mining: Overburden aquifer protection. Chemosphere. (â…±) Effects of height/diameter ratio on failure and damage properties of granite under coupled bending and splitting deformation. Engineering Fracture Mechanics.
